# Distance-Based Propagation for Efficient Knowledge Graph Reasoning

**Harry Shomer[1]    Yao Ma[2]    Juanhui Li[1]    Bo Wu[3]**
**Charu C. Aggarwal[4]    Jiliang Tang[1]**

[1] Michigan State University    [2] Rensselaer Polytechnic Institute    [3] Colorado School of Mines
[4] IBM T. J. Watson Research Center

`{shomerha, lijuanh1, tangjili}@msu.edu; may13@rpi.edu`
`bwu@mines.edu; charu@us.ibm.com`

## Abstract

Knowledge graph completion (KGC) aims to predict unseen edges in knowledge graphs (KGs), resulting in the discovery of new facts. A new class of methods have been proposed to tackle this problem by aggregating path information. These methods have shown tremendous ability in the task of KGC. However they are plagued by efficiency issues. Though there are a few recent attempts to address this through learnable path pruning, they often sacrifice the performance to gain efficiency. In this work, we identify two intrinsic limitations of these methods that affect the efficiency and representation quality. To address the limitations, we introduce a new method, TAGNet, which is able to efficiently propagate information. This is achieved by only aggregating paths in a fixed window for each source-target pair. We demonstrate that the complexity of TAGNet is independent of the number of layers. Extensive experiments demonstrate that TAGNet can cut down on the number of propagated messages by as much as $90\%$ while achieving competitive performance on multiple KG datasets [1].

## 1 Introduction

Knowledge graphs (KGs) encode facts via edges in a graph. Because of this, one can view the task of predicting unknown edges (i.e. link prediction) as analogous to uncovering new facts. This task is referred to as knowledge graph completion (KGC) and has attracted a bevy of research over the past decade (Bordes et al., 2013; Trouillon et al., 2016; Schlichtkrull et al., 2018; Zhu et al., 2021). Most work has focused on learning quality representations for all nodes (i.e. entities) and edge types (i.e. relations) in the graph to facilitate KGC.

Recently, methods (Zhu et al., 2021; Sadeghian et al., 2019; Zhang and Yao, 2022), have been introduced that move away from the embedding-based approach and focus instead on learning directly from path-based information. One recent GNN-based method, NBFNet (Zhu et al., 2021), draws inspiration from the Bellman-Ford algorithm by computing path information through dynamic programming. By doing so, it learns pairwise embeddings between all node pairs in an inductive fashion. It achieves state-of-the-art performance in both the transductive and inductive KGC settings. In this work, we refer to such methods as path-based GNNs. However, a downside of path-based GNNs is their inefficiency. This limits their ability in large real-world graphs. Furthermore, it inhibits their ability to propagate deeply in the graph. Two recent methods have been proposed to address the inefficiency problem, i.e., A$^*$Net (Zhu et al., 2022) and AdaProp (Zhang et al., 2023), by only propagating to a subset of nodes every iteration. However, they still tend to propagate unnecessary and redundant messages.

For path-based GNNs, only the source node is initialized with a non-zero message at the beginning of the propagation process. Such models often run a total of $T$ layers, where, in each layer, all nodes aggregate messages from their neighboring edges. We identify that this design is inefficient by making the following two observations. (1) **Empty Messages:** In the propagation process, a node only obtains non-empty messages when the number of propagation layers is $\geq$ the shortest path distance between the source and the node. This means that a large number of nodes far from the source node only aggregate "empty" messages in the early propagation layers. Nonetheless, path-based GNN models such as NBFnet propagate these unnecessary "empty messages" in these early propagation layers. (2) **Redundant Messages:** To ensure path information from the source reach distant nodes, the number of layers $T$ needs to be sufficiently large. However, a large $T$ induces the propagation of redundant messages for those nodes that

---

[1]The code is available at `https://github.com/HarryShomer/TAGNet`

are close to the source node. Intuitively, short paths contain more significant information than long ones (Katz, 1953). The "close" nodes typically aggregate enough information from shorter paths in the early propagation layers. Propagating messages for longer paths in later layers for "close" nodes does not provide significant information and needlessly adds to the complexity. More details on these two observations are provided in Section 3.1.

To address these limitations and make the propagation process more efficient, we aim to develop an algorithm that limits the propagation of "empty" and "redundant" messages. In particular, we propose a new method **TAGNet** - **T**runc**A**ted propa**G**ation **Net**work. TAGNet only aggregates paths in a fixed window for each source-target pair, which can be considered a form of path pruning. Our contributions can be summarized as follows:

- We propose a new path-based GNN, TAGNet, which customizes the amount of path-pruning for each source-target node pair.
- We demonstrate that the complexity of TAGNet is independent of the number of layers, allowing for efficient deep propagation.
- Extensive experiments demonstrate that TAGNet reduces the number of aggregated messages by up to $90\%$ while matching or even slightly outperforming NBFNet on multiple KG benchmarks.

## 2 Preliminary

In this section, we first introduce the notation used throughout the paper. We then introduce the path formulation from Zhu et al. (2021), the generalized Bellman-Ford algorithm (Baras and Theodorakopoulos, 2010), and NBFNet (Zhu et al., 2021).

### 2.1 Notations

We denote a KG as $\mathcal{G} = \{\mathcal{V}, \mathcal{R}, \mathcal{E}\}$ with entities $\mathcal{V}$, relations $\mathcal{R}$, and edges $\mathcal{E}$. An edge is denoted as a triple and is of the form $(s, q, o)$ where $s$ is the subject, $q$ the query relation, and $o$ the object. for an incomplete fact $(s, q, ?)$. In such a problem, we refer to the node entity $s$ as the source node and any possible answer ? as the target node. Lastly, we denote the shortest path distance between nodes $s$ and $o$ as $\text{dist}(s, o)$. We assume an edge weight of 1 since KGs typically don't contain edge weights.

### 2.2 Path Formulation

Zhu et al. (2021) introduce a general path formulation for determining the existence of an edge

$(s, q, o)$. They consider doing so by aggregating all paths between $s$ and $o$, conditional on the query $q$. We denote the maximum path length as $T$ (in their paper they set $T = \infty$), $P_{s,o}^t$ represents all paths of length $t$ connecting nodes $s$ and $o$, and $\mathbf{w}_q(e_i)$ is the representation of an edge $e_i$ conditional on the relation $q$. The representation of an edge $(s, q, o)$ is given by $\mathbf{h}_q(s, o)$:

$$\mathbf{h}_q(s, o) = \bigoplus_{t=1}^{T} \bigoplus_{p \in P_{s,o}^t} \bigotimes_{i=1}^{|p|} \mathbf{w}_q(e_i). \qquad (1)$$

(Zhu et al., 2021) show that this formulation can capture many existing graph algorithms including the Katz index (Katz, 1953), Personalized PageRank (Page et al., 1999) and others.

### 2.3 Generalized Bellman-Ford

Due to the exponential relationship between path length and the number of paths, calculating Eq. (1) for large $T$ is unfeasible. As such, Zhu et al. (2021) instead model Eq. (1) via the generalized Bellman-Ford algorithm (Baras and Theodorakopoulos, 2010) which recursively computes such path information in a more efficient manner. It is formulated as:

$$\mathbf{h}_q^{(0)}(s, o) = \mathbf{1}_q(s = o), \qquad (2)$$

$$\mathbf{h}_q^{(t)}(s, o) = \left( \bigoplus_{(x,r,o) \in \mathcal{E}(o)} \mathbf{h}_q^{(t-1)}(s, x) \otimes \mathbf{w}_q(x, r, o) \right)$$
$$\oplus \mathbf{h}_q^{(0)}(s, o), \qquad (3)$$

where $\mathcal{E}(o)$ represents all edges with $o$ as the object entity, i.e., $(*, *, o)$. Zhu et al. (2021) prove that $T$ iterations of the generalized Bellman-Ford is equal to Eq. (1) with a max path length of $T$.

### 2.4 NBFNet

Zhu et al. (2021) extend Eq. (2) via the inclusion of learnable parameters. $\mathbf{w}_q(x, r, o)$ is replaced with a learnable embedding $\mathbf{w}_q(r)$ for each relation $r$. A linear transformation is further included in the aggregation. It is formulated as the following where for convenience we set $\mathbf{h}_q^{(t)}(s, o) = \mathbf{h}_o^{(t)}$ and $\mathbf{w}_q(x, r, o) = \mathbf{w}_q(r)$:

$$\mathbf{h}_o^{(0)} = \text{INDICATOR}(u, v, q),$$
$$\mathbf{h}_o^{(t)} = \text{AGG}\Big( \big\{ \text{MSG}(\mathbf{h}_x^{(t-1)}, \mathbf{w}_q(r)) \mid \qquad (4)$$
$$(x, r, o) \in \mathcal{E}(o) \big\} \cup \{\mathbf{h}_o^{(0)}\} \Big).$$

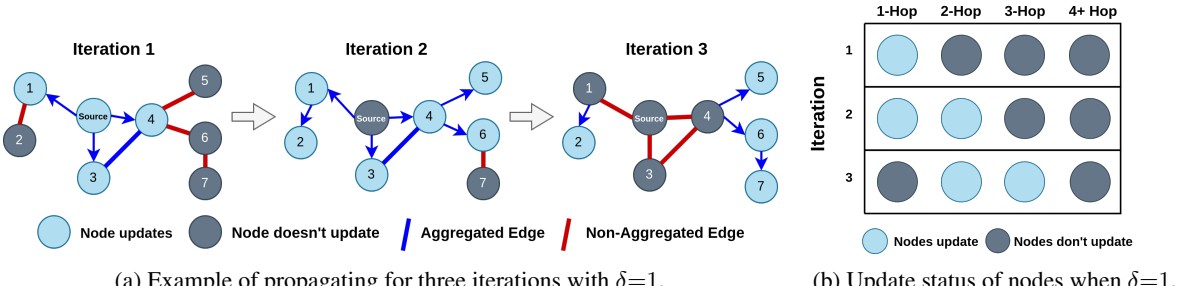

(a) Example of propagating for three iterations with $\delta$=1.

(b) Update status of nodes when $\delta$=1.

Figure 1: Example of our algorithm when $\delta = 1$. We note that an **undirected** blue edge indicates that both nodes aggregate each other. A **directed** edge indicates that only the head node aggregates the tail node. E.g., at iteration 2 node 2 aggregates node 1, however node 1 doesn't aggregate node 2.

The representation of the source node $\mathbf{h}_s^{(0)}$ is initialized to a learnt embedding, $\mathbf{q}_r$, corresponding to the query relation $r$. For all other nodes ($o \neq s$), they learn a separate initial embedding. However in practice they simply initialize the other nodes to the $\mathbf{0}$ vector. For the AGG function they consider the sum, max, min and PNA operations. For the MSG function they consider the TransE (Bordes et al., 2013), DistMult (Yang et al., 2015), and RotatE (Sun et al., 2019) operators. The final representation is passed to a score function $f$ which is modeled via an MLP.

## 3 The Proposed Framework

In this section, we propose a new approach to improve the efficiency of path-based GNN models. Inspired by two observations in Section 3.1, we proposed a simple but effective distance-based pruning strategy. We then introduce a truncated version of the generalized Bellman-Ford algorithm that achieves the goal of our proposed pruning strategy. Finally, we describe a neural network model based on the truncated Bellman-Ford.

### 3.1 Motivation

In this subsection, we discuss the motivation behind our framework design. In particular, we suggest that the inefficiency of path-based GNNs is mainly due to two observations: (1) the aggregation of many empty messages and (2) the proliferation of redundant messages when the number of layers is large. Next, we detail our observations and how they inspire us to design a more efficient method.

**Observation #1**: Empty Messages. Most path-based GNNs aggregate empty messages that do not contain any path information. This has the effect of increasing the model complexity without

any obvious benefit. We provide an illustrative example. In Figure 1a, during the first iteration, node 7 will try to aggregate path information from node 6. However, all node representations, outside of the source, are initialized to zero ("empty messages"). Hence, a non-informative "empty message" will be passed to node 7 from node 6. In fact, in the first iteration, only the 1-hop neighbors of the source aggregate non-empty messages which contains information on paths with length 1. Only after two iterations will node 6 contain path information from the source. Therefore aggregating any messages before the third iteration will not lead to any path information for node 7. However, both NBFNet (Zhu et al., 2021) and A*Net (Zhu et al., 2022) will aggregate such messages, leading to increased complexity without any gain in additional path information. This observation suggests that *a node o of distance $dist(s, o)$ from the source can only aggregate path information from iteration $t = dist(s, o)$ onwards.*

**Observation #2**: Redundant Messages. Due to their design, *path-based GNNs with $T$ layers can only learn representations for nodes within $T$ hops of the source node*. However, since the time complexity of all existing methods is proportional to the number of layers, learning representations for nodes far from the source (i.e., distant nodes) can be very inefficient. In particular, as we discussed in Section 1, this mainly afflicts *target nodes closer to the source*. Again, we utilize Figure 1a for illustration. In the first two iterations the node 4 aggregates two paths including (source, 4) and (source, 3, 4). These paths provide significant information between the source and 4. Comparatively, in the 6-th iteration node 4 aggregates paths[2] of

---

[2] Strictly, these walks are not paths, as they contain re-

length 6, which reach further nodes and return to node 4. Since these paths already contain information present in shorter paths, little information is gained by aggregating them. Our empirical study in Section 4.3 also verifies that aggregating paths of longer length relative to the target node have little to no positive effect on performance.

These two observations suggest that the efficiency of path-based GNN methods is low when there are nodes of diverse distances to the source. We verify this by analyzing the distance distribution for all test samples on the WN18RR (Dettmers et al., 2018) dataset. For each sample we calculate the shortest path distance between both nodes and plot the distribution of the distances over all samples. The results are shown in Figure 2. We note that around 25% of samples have a shortest distance $\geq 5$. To aggregate information for these distant nodes, it is necessary to set $T$ to $\geq 5$. In this case, nodes of larger distance will propagate empty messages for the first few iterations (Observation 1). Furthermore, about 35% of the samples have a shortest distance of 1. Such samples will aggregate redundant messages after a few iterations (Observation 2).

**Our Design Goal**: The key to improving the efficiency of path-based GNNs is to modify their aggregation scheme. In particular, based on the aggregation scheme of path-based GNNs, all target nodes are aggregating paths with lengths ranging from 1 to $T$. Such paths contain many empty and redundant messages. To reduce the aggregation of those non-informative messages, we propose to customize the aggregations for each target node. Specifically, for close nodes, we do not aggregate long paths as they are redundant. For distant nodes, we do not aggregate short paths as they are empty. As such, we customize the aggregation process for each target node according to its distance from the source. Based on this intuition, we reformulate the path formulation, Eq. (1), as follows.

$$\mathbf{x}_q(s, o) = \bigoplus_{t=\text{dist}(s,o)}^{\text{dist}(s,o)+\delta} \bigoplus_{p \in P_{s,o}^t} \bigotimes_{i=1}^{|p|} w(e_i), \quad (5)$$

where $\delta \geq 0$ is an offset. The parameter $\delta$ can be considered as a form of path pruning as it controls the paths we aggregate relative to the shortest path

peated nodes and edges. In this paper, we follow the convention of the path-based GNN papers to loosely call them paths.

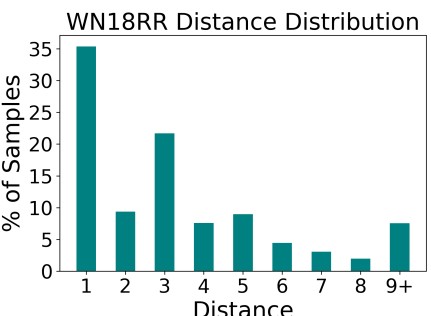

Figure 2: Test Distance Distribution for WN18RR

distance. For example, when $\delta = 0$, it only aggregates those paths of the shortest distance for all node pairs. Empirical observations in Section 4.3 validate our use of pruning based on an offset $\delta$.

Due to the high complexity of Eq. (5), it is not practical to directly calculate it. Hence, based on the generalized Bellman-Ford algorithm (Baras and Theodorakopoulos, 2010), we propose a truncated version of the Bellman-Ford algorithm for calculating Eq. (5) in a more efficient fashion.

### 3.2 Truncated Bellman-Ford

From our design goal, we are interested in capturing all paths of length $\text{dist}(s, o) \leq l \leq \text{dist}(s, o) + \delta$. To achieve this goal, for node $o$, we begin aggregating at iteration $t = \text{dist}(s, o)$ and stop aggregation after iteration $t = \text{dist}(s, o) + \delta$. This helps avoid aggregating empty messages before $\text{dist}(s, o)$-th iteration and redundant messages after $\text{dist}(s, o) + \delta$ iterations. However, during the iterations between $\text{dist}(s, o)$ and $\text{dist}(s, o) + \delta$, there are still potential empty messages. For example, any node $v$ with the shortest distance to source larger than $\text{dist}(s, o) + \delta$ always contains empty messages during these iterations. Hence, to further avoid aggregating empty messages, we only allow aggregation from a subset of the neighboring nodes of $o$. More formally, we formulate the above intuition into the following constrained edge set $\mathcal{C}(s, o, t)$ through which node $o$ aggregates information at iteration $t$.

$$\mathcal{C}(s, o, t) = \begin{cases} \emptyset, \text{ if } t < \text{dist}(s, o) \text{ or} \\ \quad t > \text{dist}(s, o) + \delta \\ \{(v, r, o) \in \mathcal{E}(o) \mid \\ \quad \text{dist}(s, v) < \text{dist}(s, o) + \delta\}, \text{else} \end{cases} \quad (6)$$

Based on this constraint set of edges for node $o$, we update the generalized Bellman-Ford algorithm

(Eq. 2) as follows where $\mathcal{C} = \mathcal{C}(s, o, t)$:

$$\mathbf{x}_q^{(t)}(s, o) = \left( \bigoplus_{(v, r, o) \in \mathcal{C}} \mathbf{x}_q^{(t-1)}(s, v) \otimes \mathbf{w}_q(v, r, o) \right)$$
$$\oplus \mathbf{x}_q^{(0)}(s, o). \tag{7}$$

The following theorem shows that the aggregation scheme proposed in Eq. (7) results in aggregation of the correct paths as described in Eq. (5).

**Theorem 1.** *Given a source node $s$, query $q$, and target node $o$, the final representation, $\mathbf{x}_q^F(s, o)$ only aggregates all path representations whose path length is between $dist(s, o)$ and $dist(s, o) + \delta$ for all $o \in V$. It therefore contains all information present in Eq. (5) such that,*

$$\mathbf{x}_q^F(s, o) = \bigoplus_{t = dist(s, o)}^{dist(s, o) + \delta} \bigoplus_{p \in P_{s, o}^t} \bigotimes_{i=1}^{|p|} w(e_i). \tag{8}$$

The detailed proof of Theorem 1 is provided in Appendix A. This design has the following advantages. **(1)** We don't begin aggregating messages until layer $t = dist(s, o)$. This helps avoid the aggregation of many empty messages for nodes far from the source. **(2)** We stop aggregating messages at layer $t = dist(s, o) + \delta$. This ensures that for close nodes we don't aggregate many redundant messages. Furthermore, it ensures that we will always aggregate paths of $\delta + 1$ different lengths for all target nodes regardless of their distance from the source. **(3)** In Section B.2, we demonstrate that the complexity of this design is *independent of the number of layers*, allowing for deep propagation.

**An Illustrative Example.** We given an example of the effect of constraints on propagation in Figure 1 where $s$ = source. Figure 1a shows the involved nodes and edges over three iterations when $\delta = 1$. We observe that only a portion of the nodes and edges are involved at any one iteration. For example, at iteration 1 only the 1-hop neighbors and the edges connecting them to the source are involved. This is because they are the only nodes and edges able to receive any path information at that stage. Figure 1b details the update status of nodes by distance from the source node. We note how as the iteration increases the number of nodes updated shift to the right in groups of two. Furthermore since we only iterate for three iterations, the 4+ hop neighbors never

update as there is no available path information for them until iteration 4.

### 3.3 Degree Messages

An effect of pruning paths, especially with low $\delta$, is that it can lead to very few messages being aggregated. This is especially true for smaller or sparser graphs. One consequence of few messages being aggregated is that it can make it difficult for a node to discern the properties of its neighborhood (e.g. degree). We give an example of node 4 in Figure 1. For each of the first 2 iterations, it only aggregates messages from 2/4 of it's neighbors. As such, it never aggregates messages from all its neighbors at the same iteration. This can lead to a failure of node 4 to properly discern it's degree, as the number of non-empty messages in each iteration is only a portion of the overall degree. Since the degree is known to be an important factor in link prediction (Newman, 2001; Adamic and Adar, 2003), we want to preserve the degree information for all nodes.

In order to preserve the degree information for each node, we consider encoding the degree via the use of pseudo messages. Specifically, we want to add enough messages such that the total number of messages aggregated for a node $o$ is equivalent to its degree. We refer to such messages as *degree messages*. Going back to our example in Figure 1, for node 4 at iteration 1 and 2 we would add 2 degree messages so that the total number of messages is 4. Formally, we denote the degree of a node $o$ as $b_o$. The number of messages to add at iteration $t$ is given by $\rho_o = b_o - |C(s, o, t)|$.

For the value of the messages, we learn a separate embedding denoted as $\mathbf{x}_{\text{deg}}^{(t)}$ that is the same across all nodes. Since the value of each message is the same we can avoid explicitly aggregating each degree message individually. Instead, we just aggregate one message that is equal to the number of degree messages multiplied by the degree embedding,

$$\mathbf{x}_{\text{deg}}^{(t)}(s, o) = \rho_o \cdot \mathbf{x}_{\text{deg}}^{(t)}, \tag{9}$$

where $\mathbf{x}_{\text{deg}}^{(t)}(s, o)$ is the value of the degree message for node $o$ at iteration $t$. This edge is then added to the set of messages to be aggregated, $C(s, o, t)$. Since this is equivalent to computing and aggregating only one edge, it has no effect on the model complexity. Experimental results in Section 4.4 validate the effectiveness of degree messages.

## 3.4 GNN Formulation

We follow similar conventions to NBFNet when converting Eq. (6) and Eq. (7) to a GNN. We denote the embedding of a source node $s$ and arbitrary target node $o$ as $\mathbf{x}_q(s, o)$. We further represent the indicator query embeddings as $\mathbf{x}_q$ and the layer-wise relation embeddings as $\mathbf{x}_r^{(t)}$.

We utilize the INDICATOR function described in Section 2.4, PNA (Corso et al., 2020) for the AGGREGATE function, and DistMult (Yang et al., 2015) for the MSG function. The probability of a link existing between a source-target pair is determined via a score function $f$. Both the final representation of the pair and the query embedding are given as input. The output of $f$ is then passed to a sigmoid to produce a probability,

$$p(s, o) = \sigma \left( f \left( \mathbf{x}_q^{\mathrm{F}}(s, o), \mathbf{x}_q \right) \right), \qquad (10)$$

where $\mathbf{x}_q^{\mathrm{F}}(s, o)$ is the final pair representation. The full algorithm is detailed in Appendix B.1. We run a total of $T$ layers. We further show in in Appendix B.2 that time complexity is independent of the number of layers. This enables TAGNet to propagate for more layers than existing path-based GNNs.

Furthermore, due to its general design, TAGNet can also be integrated with other efficiency-minded methods like A*Net. This is described in more detail in Appendix B.3. Extensive experiments in Sections 4.1 and 4.2 also demonstrate that combining both methods can significantly reduce the number of messages propagated by A*Net without sacrificing performance.

## 3.5 Target-Specific $\delta$

A drawback of our current design is that we assume a single offset $\delta$ for all possible node pairs. However, for some pairs we may want to consider propagating more or less iterations. For example, in Figure 1 we may only want to consider $\delta = 0$ for the target node 2 due to the limited number of paths connecting it to the source. However for node 4, which is concentrated in a denser portion of the subgraph, we may want to consider a higher value of $\delta$ such as 1 or 2 to capture more path information. We next detail our method for achieving this.

### 3.5.1 Target-Specific $\delta$ via Attention

A target-specific $\delta$ can be attained by realizing the connection between the hidden representations and

the value of $\delta$. Let's denote the value of the hyperparameter $\delta$ as $\hat{\delta}$. For a source-target node pair $(s, o)$, we only aggregate paths from length $\mathrm{dist}(s, o)$ to $\mathrm{dist}(s, o) + \hat{\delta}$. At iteration $t = \mathrm{dist}(s, o)$ we aggregate paths of length $\mathrm{dist}(s, o)$ and at iteration $t = \mathrm{dist}(s, o) + 1$ only those paths of length $\mathrm{dist}(s, o) + 1$, and so on until $t = \mathrm{dist}(s, o) + \hat{\delta}$. The set of hidden representations for a node pair is as follows where for convenience we represent $\mathbf{x}_q(s, o)$ as $\mathbf{x}_{(s,o)}$:

$$\mathrm{Hiddens}(s, o) = \left[ \mathbf{x}_{(s,o)}^{\mathrm{dist}(s,o)}, \cdots, \mathbf{x}_{(s,o)}^{(\mathrm{dist}(s,o)+\hat{\delta})} \right]. \qquad (11)$$

The first hidden representation only contains paths of shortest length and therefore corresponds to $\delta = 0$. Since the paths accumulate over hidden representations via a self-loop, $\mathbf{x}_{(s,o)}^{(\mathrm{dist}(s,o)+1)}$ contains all paths of length $\mathrm{dist}(s, o)$ and $\mathrm{dist}(s, o) + 1$, corresponding to $\delta = 1$. As such, the final hidden representation is equivalent to $\delta = \hat{\delta}$. Therefore, choosing a target-specific $\delta$ is achieved by selecting one of the hidden representations as the final representation.

We utilize attention to determine which value of $\delta$ is best for a specific target node. This is formulated as the following:

$$\mathbf{x}_{(s,o)}^{\mathrm{F}} = \sum_{\delta=0}^{\hat{\delta}} \alpha_{(s,o)}^{\delta} \mathbf{x}_{(s,o)}^{(\mathrm{dist}(s,o)+\delta)}, \qquad (12)$$

where $\alpha_{(s,o)}^{\delta}$ is the corresponding attention weight for the hidden representation $\mathbf{x}_{(s,o)}^{(\mathrm{dist}(s,o)+\delta)}$. For each possible value of $\delta$, $\alpha_{(s,o)}^{\delta}$ is given by:

$$\tilde{\alpha}_{(s,o)}^{\delta} = g \left( \mathbf{x}_{(s,o)}^{(\mathrm{dist}(s,o)+\delta)}, \mathbf{x}_q \right)$$
$$\alpha_{(s,o)}^{\delta} = \mathrm{Softmax}(\tilde{\alpha}_{(s,o)}^{\delta}).$$

We model $g$ as an MLP that takes both the hidden representation and the query embedding as input. Taking inspiration from A*Net (Zhu et al., 2022), we conjecture that a well-learned score function can help determine which representations are better than others. As such, we further consider modeling $g$ as its own function or having it share parameters with the score function $f$, Eq. (10). Lastly, we show in Appendix B.2 that the time complexity is unchanged when using a target-specific $\delta$.

## 4 Experiment

In this section, we evaluate the effectiveness of our proposed framework on KGC under both the trans-

Table 1: Transductive Results. Best results are in **bold** and the 2nd best underlined.

| Method Type | Method | FB15k-237 | | | WN18RR | | |
|---|---|---|---|---|---|---|---|
| | | MRR | Hits@1 | Hits@10 | MRR | Hits@1 | Hits@10 |
| **Embeddings** | TransE | 0.294 | - | 0.465 | 0.226 | - | 0.501 |
| | DistMult | 0.241 | 0.155 | 0.419 | 0.43 | 0.39 | 0.49 |
| | ComplEx | 0.247 | 0.158 | 0.428 | 0.44 | 0.41 | 0.51 |
| **GNNs** | R-GCN | 0.273 | 0.182 | 0.456 | 0.402 | 0.345 | 0.494 |
| | CompGCN | 0.355 | 0.264 | 0.535 | 0.479 | 0.443 | 0.546 |
| **Path-Based** | DRUM | 0.343 | 0.255 | 0.516 | 0.486 | 0.425 | 0.586 |
| | RED-GNN | 0.374 | 0.283 | 0.558 | 0.533 | 0.485 | 0.624 |
| | AdaProp | 0.392 | 0.309 | 0.555 | 0.553 | 0.502 | 0.652 |
| | NBFNet | 0.415 | 0.321 | 0.599 | 0.551 | 0.497 | **0.666** |
| | A*Net | 0.414 | 0.324 | 0.592 | 0.547 | 0.490 | 0.658 |
| **TAGNet** | + A*Net | 0.409 | 0.323 | 0.577 | 0.555 | 0.502 | 0.657 |
| | Fixed $\delta$ | **0.421** | **0.328** | **0.602** | 0.562 | 0.509 | **0.667** |
| | Specific $\delta$ | 0.417 | **0.328** | 0.592 | **0.565** | **0.513** | **0.667** |

Table 2: Inductive Results (evaluated with Hits@10). Ours results are averaged over 5 runs

| Method | FB15k-237 | | | | WN18RR | | | |
|---|---|---|---|---|---|---|---|---|
| | v1 | v2 | v3 | v4 | v1 | v2 | v3 | v4 |
| NeuralLP | 0.468 | 0.586 | 0.571 | 0.593 | 0.772 | 0.749 | 0.476 | 0.706 |
| DRUM | 0.474 | 0.595 | 0.571 | 0.593 | 0.777 | 0.747 | 0.477 | 0.702 |
| GraIL | 0.429 | 0.424 | 0.424 | 0.389 | 0.760 | 0.776 | 0.409 | 0.687 |
| RED-GNN | 0.483 | 0.629 | 0.603 | 0.621 | 0.799 | 0.780 | 0.524 | 0.721 |
| AdaProp | 0.470 | 0.651 | 0.620 | 0.614 | 0.798 | **0.836** | **0.582** | 0.732 |
| NBFNet | **0.607** | **0.704** | 0.667 | **0.668** | **0.826** | 0.798 | 0.568 | 0.694 |
| A*Net | 0.535 | 0.638 | 0.610 | 0.630 | 0.810 | 0.803 | 0.544 | 0.743 |
| TAGNet + A*Net | 0.541 | 0.646 | 0.604 | 0.623 | 0.813 | 0.805 | 0.535 | **0.745** |
| TAGNet (fixed $\delta$) | 0.596 | 0.700 | **0.677** | 0.666 | 0.816 | 0.796 | 0.534 | 0.734 |
| TAGNet (specific $\delta$) | 0.596 | 0.698 | 0.675 | 0.661 | 0.818 | 0.803 | 0.544 | 0.737 |

ductive and inductive settings. We also empirically analyze the efficiency and conduct ablation studies on each component. The experimental details are listed in Appendix C. We note that for a fair comparison between path-based GNNs, we run each model using 6 layers and a hidden dimension of 32 as is done in both (Zhu et al., 2021) and (Zhu et al., 2022). Please see Appendix C.2 for more details.

## 4.1 Effectiveness of TAGNet

In this subsection, we present the results of TAGNet compared with baselines on both transductive and inductive settings. We further detail the results when combining TAGNet with A*Net.

**Transductive Setting**: The results on the transductive setting are shown in Table 1. We observe that TAGNet achieves strong performance with just a fixed $\delta$. In particular, it outperforms A*Net and AdaProp on most metrics. Also compared to NBFnet, which doesn't utilize pruning, TAGNet achieves comparable or even stronger performance. This indicates that the proposed pruning strategy

mostly reduces redundant aggregations that do not impair the models effectiveness.

**Inductive Setting**: Table 2 shows the results on the inductive setting. TAGNet achieves strong performance on both datasets. In particular, it achieves comparable performance to the non-pruning version of NBFNet. Furthermore, TAGNet significantly outperforms A*Net and AdaProp on the FB15k-237 splits, demonstrating the advantage of the proposed pruning strategy.

**TAGNet + A*Net:** We further test combining the pruning strategy of both TAGNet and A*Net together (see Appendix B.3 for more details). Compared to A*Net, we observe that TAGNet+A*Net achieves comparable if not better performance under all settings despite aggregating much fewer messages (see subsection 4.2). This suggests that the pruning strategy in A*Net fails to prune many irrelevant paths, allowing TAGNet to work complementary to it.

## 4.2 Efficiency of TAGNet

In this subsection, we empirically evaluate the efficiency of our model against NBFNet. Specifically, we compare the mean number of messages aggregated per sample during training.

Figure 3 shows the % decrease in the number of messages of TAGNet as compared to NBFNet. All models are fit with 6 layers. We observe two trends. The first is that both FB15k-237 datasets follow a similar relationship that is close to what's expected of the worst-case complexity detailed in Appendix B.2. On the other hand, the WN18RR datasets pass much fewer messages as they hover above 90% for all $\delta$. This is likely because WN18RR is a very sparse graph. This gives TAGNet plenty of opportunities to prune paths.

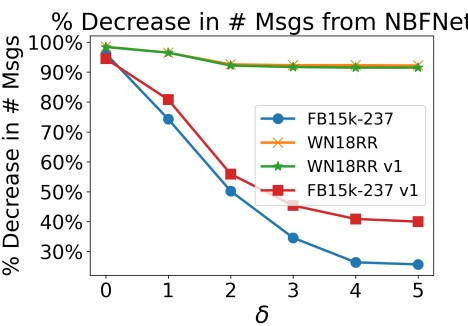

Figure 3: % Decrease in NBFNet Messages

We further compare the efficiency of just A\*Net and A\*Net + TAGNet. As before, we calculate the total number of messages passed for both methods. We fix $\delta = 2$. Table 3 show the % decrease in the number of messages when utilizing both techniques compared to just A\*Net. We observe a large reduction in both the inductive and transductive setting. Since the performance of A\*Net + TAGNet is on par with just A\*Net, it suggests that A\*Net fails to prune many unneeded messages that do not improve performance. Furthermore, we find that the reduction in the number of messages becomes more pronounced with more layers, suggesting that TAGNet is even more useful when deep propagation is necessary.

## 4.3 Effect of $\delta$

In this subsection, we evaluate the effect of the offset $\delta$ on TAGNet test performance (w/o the target-specific setting). We fix the number of layers at 6 and vary $\delta$ from 0 to 5. We report results for both the transductive and inductive settings in Figures 4 and 5, respectively. For the inductive setting, we

Table 3: % Decrease in # Msgs for A\*Net vs. A\*Net + TAGNet

| Dataset | 6 Layers | 7 Layers | 8 Layers |
|---|---|---|---|
| FB15k-237 | 39% | 51% | 59% |
| FB15k-237 v1 | 30% | 44% | 66% |
| WN18RR | 10% | 17% | 26% |
| WN18RR v1 | 25% | 37% | 46% |

chose version v1 of both datasets as the representative datasets. For both transductive datasets, we find that the performance plateaus at $\delta = 2$. A similar trend is observed for FB15k-237 v1. Interestingly, for WN18RR v1, the performance is constant when varying $\delta$. This suggests that for some datasets almost all of the important information is concentrated in paths of the shortest length.

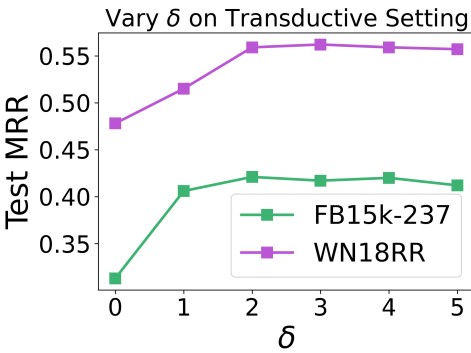

Figure 4: Performance varying $\delta$ on Transductive setting.

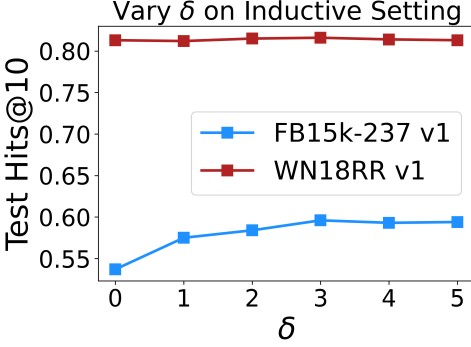

Figure 5: Performance varying $\delta$ on Inductive setting.

## 4.4 Effect of Degree Messages

We demonstrate the effect of the degree messages described in Section 3.3. Table 4 shows the performance of TAGNet when trained with and without degree messages. We report the performance on all of the inductive splits for both FB15k-237 and WN18RR. Interestingly, we observe that while there is a consistent gain on FB15k-237, it often

hurts performance on WN18RR. This may imply that preserving the degree information of each node is more important on FB15k-237 than WN18RR.

Table 4: Effect of Degree Messages on Inductive Splits

| Dataset | Split | w/o Msgs | with Msgs |
|---------|-------|----------|-----------|
| FB15k-237 | V1 | 0.594 | **0.596** |
|           | V2 | 0.684 | **0.698** |
|           | V3 | 0.653 | **0.675** |
|           | V4 | 0.648 | **0.661** |
| WN18RR | V1 | 0.815 | **0.818** |
|        | V2 | **0.803** | 0.781 |
|        | V3 | **0.544** | 0.465 |
|        | V4 | **0.737** | 0.718 |

## 5 Related Work

We give a brief overview of different types of KGC methods. **(1) Embedding-Based Methods**: Such methods are concerned with modeling the interactions of entity and relation embeddings. TransE (Bordes et al., 2013) models each fact as translation in the embedding space while DistMult (Yang et al., 2015) scores each fact via a bilinear diagonal function. ComplEx (Trouillon et al., 2016) extends DistMult by further modeling the embeddings in the complex space. Lastly, Nodepiece (Galkin et al., 2021) attempts to improve the efficiency of embedding-based KGC methods by representing each entity embedding as a combination of a smaller set of subword embeddings. Since this method concerns embedding-based techniques, it is orthogonal to our work. **(2) GNN-Based Methods**: GNN methods extend traditional GNNs by further considering the relational information. CompGCN (Vashishth et al., 2019) encodes each message as a combination of neighboring entity-relation pairs via the use of compositional function. RGCN (Schlichtkrull et al., 2018) instead considers a relation-specific transformation matrix to integrate the relation information. **(3) Path-Based Methods**: Path-based methods attempt to leverage the path information connecting two entities to perform KGC. NeuralLP (Yang et al., 2017) and DRUM (Sadeghian et al., 2019) learn to weight different paths by utilizing logical rules. More recently, NBFNet (Zhu et al., 2021) considers path information by learning a parameterized version of the Bellman-Ford algorithm. A similar framework, RED-GNN (Zhang and Yao, 2022) also attempts to take advantage of dynamic programming to aggregate path information. Both A*Net (Zhu et al.,

2022) and AdaProp (Zhang et al., 2023) attempt to prove upon the efficiency of the previous methods by learning which nodes to propagate to.

## 6 Conclusion

In this paper we identify two intrinsic limitations of path-based GNNs that affect the efficiency and representation quality. We tackle these issues by introducing a new method, TAGNet, which is able to efficiently propagate path information. This is realized by only aggregating paths in a fixed window for each source-target pair. We demonstrate that the complexity of TAGNet is independent of the number of layers. For future work, we plan on exploring methods to capture path information without having to perform a separate round of propagation for every individual source node.

## Limitations

Our work has a couple of limitations. One is that it our study is limited to only knowledge graph completion. This excludes non-relational link prediction tasks. Future work can ascertain the effectiveness of TAGNet on other types of link prediction. Second, all path-based GNNs still require to propagate from each source-relation pair individually. This can pose a significant bottleneck when many samples need to be tested. We plan on exploring methods to capture path information without having to propagate for each individual pair.

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

## A   Proof Details of Theorem 1

We prove Theorem 1 via induction on the path length $l$. We denote all nodes a distance $l$ from the source node $s$ as $V_s^l$. The path length offset is represented by $\delta$. Lastly, for convenience we split the constraints in Eq. (6) into two: a *node constraint* and an *edge constraint*. We formulate it as the following where $\text{Node}_\delta(s, o, t)$ represents the node constraint and $\text{EdgeC}_\delta(s, o, u)$ the edge constraint:

$$\text{Node}_\delta(s, o, t) = t - \delta \leq \text{dist}(s, o) \leq t, \tag{13}$$

$$\text{EdgeC}_\delta(s, o, u) = \text{dist}(s, u) < \text{dist}(s, o) + \delta \tag{14}$$

**Base Case** ($l$=1): We want to show for all $l = 1$ hop neighbors of $s$, $o \in V_s^1$, their final representation $x_q^F(s, o)$ aggregates all path representations in the range $[0, 1 + \delta]$. To be true, the embedding $x_q^F(s, o)$ must satisfy two conditions:

1. *Condition 1*: The final embedding $x_q^F(s, o)$, contains all paths representations of length less than or equal to $1 + \delta$ between $s$ and $o$.

2. *Condition 2*: The final embedding $x_q^F(s, o)$ contains no other path information.

*Condition 1*: For it to be true, a node $o \in V_s^1$ must aggregate all edges of the form $(u, r, o)$ where $u$ belongs to the set:

$$U_{s,o}^{(0,\delta)} = \{u \mid (u, r, o) \in \mathcal{E}_o,\ u \in \{V_s^0, V_s^1, \cdots, V_s^\delta\}\}, \tag{15}$$

where $\mathcal{E}_o$ represents all edges where $o$ is the target node. It's intuitive that all paths starting at $s$ of length $\in [0, \delta + 1]$ must pass through the nodes in the set $U_{s,o}^{(0,\delta)}$ in order to reach $o$. We prove in Theorem 2 that $o$ will aggregate all nodes in the set $U_{s,o}^{(0,\delta)}$.

*Condition 2*: We want to demonstrate that the representation of node $o$ aggregates no other path information such that $x_q^{(\delta+1)}(s, o) = x_q^F(s, o)$. This is true as per the node constraint (Eq. (13)) the representation of a node $o$ stops updating after iteration $k = 1 + \delta$.

**Inductive Step**: We assume that for all m-hop neighbors of $s$, $o \in V_s^m$, their final representation $x_q^F(s, o)$ aggregates all path representations of length between $[m, m + \delta]$. This is achieved by a node $o$ aggregating all edges $(u, r, o)$ where $u$ belongs to the set:

$$U_{s,o}^{(m-1,m-1+\delta)} = \{u \mid (u, r, o) \in \mathcal{E}_o,\ u \in \{V_s^{m-1}, \cdots, V_s^{m-1+\delta}\}\}, \tag{16}$$

as all such paths must pass through these nodes. We note that this implies that:

- The set of nodes $U_{s,o}^{(m-1,m-1+\delta)}$ must themselves only contain all path representations of lengths $[m - 1, m - 1 + \delta]$ when aggregated by $o \in V_s^m$.

- The set of nodes $U_{s,o}^{(m-1,m-1+\delta)}$ must obtain such path information by iteration $k = m - 1 + \delta$. This must be true as per the node constraint $o$ will last update at iteration $k = m + \delta$.

We now want to show for all $(m + 1)$ hop neighbors of $s$, $o \in V_s^{m+1}$, their final representation $x_q^F(s, o)$ aggregates all path representations of of length between $[m + 1, m + 1 + \delta]$. This requires showing that $x_q^F(s, o)$ (1) contains all paths representations between $[m + 1, m + 1 + \delta]$ between $s$ and $o$ and (2) it contains no other path information.

*Condition 1*: For $o \in V_s^{m+1}$ to aggregate all paths of length between $m + 1$ and $m + 1 + \delta$, their representation must aggregate all edges $(u, r, o)$ where $u$ belongs to the set:

$$U_{s,o}^{(m,m+\delta)} = \{u \mid (u, r, o) \in \mathcal{E}_o,\ u \in \{V_s^m, \cdots, V_s^{m+\delta}\}\}. \tag{17}$$

Such edges are aggregated by $o \in V_s^{m+1}$ via the edge constraint. Furthermore,

- From the inductive step we know that nodes $U_{s,o}^{(m-1,m-1+\delta)} = U_{s,v}^{(m,m+\delta)} \setminus V_s^{m+\delta}$ have already aggregated all path representations of lengths $[m-1, m-1+\delta]$ by iteration $k = m + \delta$.

- From both constraints we know that $\forall u \in V_s^{m+\delta}$ will only contain all path representations of length $m + \delta$ (i.e. shortest path) by iteration $k = m + \delta$.

As such, after aggregating the nodes in the set $U_{s,o}^{(m,m+\delta)}$ the representation $x_q^{(m+\delta)}(s,u)$ will contain all paths representations between $m$ and $m + \delta$. Per the node constraint, $\forall o \in V_s^{m+1}$ last update at iteration $k = m + 1 + \delta$. Therefore by aggregating $U_{s,o}^{(m,m+\delta)}$ at iteration $k = m + 1 + \delta$, the representation $x_q^{(m+1+\delta)}(s,o)$ will contain all path representations between length $m+1$ and $m+1+\delta$.

*Condition 2*: Lastly, we want to show that $\forall o \in V_s^{m+1}$ the final representation $x_q^F(s,o)$ will only contain path representations of length $m + 1$ to $m + 1 + \delta$. This is true as per the node constraint the representation of a node $o \in V_s^{m+1}$ last updates at iteration $k = m + 1 + \delta$. Therefore $x_q^{(m+1+\delta)}(s,o) = x_q^F(s,o)$. As such, the final representation only aggregates paths of length between $m + 1$ and $m + 1 + \delta$.

**Theorem 2.** *We are given a source node $s$, query $q$, and target node $o$ which is a 1-hop neighbor of $s$. The final representation of a 1-hop neighbor $o$, $\mathbf{x}_q^F(s,o)$, will **at minimum** aggregate all path representations whose path length is between $1$ and $1 + \delta$. It therefore **at least** contains the path information,*

$$\eta = \bigoplus_{l=1}^{1+\delta} \bigoplus_{p \in P_{s,o}^l} \bigotimes_{i=1}^{|p|} w(e_i). \tag{18}$$

*This is equivalent to stating that $o$ will aggregate all nodes in the following set by iteration $k = 1 + \delta$,*

$$U_{s,o}^{(0,\delta)} = \{u \mid (u,r,o) \in \mathcal{E}_o,\ u \in \{V_s^0, V_s^1, \cdots, V_s^\delta\}\}. \tag{19}$$

We prove this Theorem via induction on the layer iteration $k$ in our algorithm 1 (denoted their as $l$).

**Base Case** ($k=1$): We want to first show that after one iteration, the representation of a 1-hop neighbor $x_q^1(s,o)$ aggregates all paths of length 1 from the source. This is achieved by $x_q^1(s,o)$ aggregating all edges connecting $o$ to $s$, i.e. $(s,r,o)$. Such edges are aggregated by $o$ as both the edge and node constraints are satisfied:

$$\text{EdgeC}_\delta(s,o,s) = 0 < 1 + \delta, \tag{20}$$
$$\text{NodeC}_\delta(s,o,1) = 1 - \delta \leq 1 \leq 1. \tag{21}$$

**Inductive Step**: We assume that at some iteration $k = n$, s.t. $n < 1 + \delta$, the representation $x_q^n(s,o)$ for $o \in V_s^1$ aggregates all path representations up to a length $n$ from the source. This is achieved by aggregating all edges that contain nodes in the set:

$$U_{s,o}^{(0,n-1)} = \{u \mid (u,r,o) \in \mathcal{E}_o,\ u \in \{V_s^0, V_s^1, \cdots, V_s^{n-1}\}\}. \tag{22}$$

Since we assume that $x_q^n(s,o)$ contains all path representations up to length $n$, then it follows that $\forall u \in U_{s,o}^{(0,n-1)}$ their corresponding representation $x_q^n(s,o)$ must also contain all paths up to length $n - 1$. As such, by node $o$ aggregating $U_{s,o}^{(0,n-1)}$ it extend the length of each path by 1.

We want to prove that at iteration $k = n + 1$, the representation $x_q^{(n+1)}(s,o)$ aggregates all path representations up to a length $n + 1$ from the source. This is achieved by aggregating all edges that contain the nodes in the set:

$$U_{s,o}^{(0,n)} = \{u \mid (u,r,o) \in \mathcal{E}_o,\ u \in \{V_s^0, V_s^1, \cdots, V_s^n\}\}. \tag{23}$$

---

**Algorithm 1** TAGNet Algorithm (fixed $\delta$)

---

**Require:**
    $s$ = Source node
    $q$ = Query relation
    $T$ = Max Number of Layers
    $\mathbf{x}$ = Embeddings
    $\delta$ = Offset
    Agg-Degree = Whether to include degree msgs

1: **Initialize:**
    $x_{(s,o)}^{(0)} = \mathbf{0}, \; \forall o \in \mathcal{V}$
    $x_{(s,o)}^{(0)} = \mathbf{x}_q$
2: **for** $t = 1...T$ **do**
3:     **for** $o \in \mathcal{V}$ **do**
4:         **if** $t - \delta \leq \text{dist}(s, o) \leq t$ **then**
5:             $\mathcal{C}(s, o, t) = \{(u, r, o) \in \mathcal{E}(o) \mid \text{dist}(s, u) < \text{dist}(s, o) + \delta\}$
6:             $\text{Msgs} = \{\mathbf{x}_{(s,u)}^{(t-1)} \odot \mathbf{x}_r^{(t)} \mid (u, r, o) \in \mathcal{C}(s, o, t)\}$

7:             **if** Agg-Degree **then**
8:                 $\rho_o = b_o - |\text{Msgs}|$
9:                 $\text{Msgs} = \text{Msgs} \cup \left\{\rho_v \cdot \mathbf{x}_{\text{deg}}^{(t)}\right\}$
10:            **end if**
11:            $\mathbf{x}_{(s,o)}^{(t)} = \text{Aggregate}\{\text{Msgs}\}$
12:         **end if**
13:     **end for**
14: **end for**
15: **return** $\mathbf{x}_{(s,o)}^{(\text{dist}(s,o)+\delta)}$ for all $o \in \mathcal{V}$

---

Per the previous inductive step, we assumed that the representations $x_q^n(s, o) \; \forall o \in V_s^n$ contain all path representations up to length $n$. Furthermore we noted that at iteration $k = n$, the representations for each node in the set $U_{s,o}^{(0,n-1)}$ must also contain all path representations up to a length $n - 1$. Since $U_{s,o}^{(0,n)} = U_{s,o}^{(0,n-1)} \cup V_s^n$, this implies that $U_{s,o}^{(0,n)}$ contain all path representations up to length $n$. Thereby when $x_q^{(n+1)}(s, o)$ aggregates the nodes in $U_{s,o}^{(0,n)}$ it aggregates all path representations up to a length $n + 1$. A node $o \in V_s^1$ will aggregate such nodes at iteration $k = n + 1$ per both constraints.

This proves by induction that for $o \in V_s^1$, their representation $x_q^{(1+\delta)}(s, o)$ aggregates all path representations of length less than or equal to $1 + \delta$.

## B    Further Details on TAGNet

### B.1    TAGNet Algorithm

The algorithm for TAGNet, with a fixed $\delta$, is presented in Algorithm 1.

### B.2    Time Complexity Analysis

Per the constraints in Eq. (6), each node can be updated at most $\delta + 1$ times and each edge can be aggregated at most $\delta + 1$ times. The shortest path distance from a source node $s$ to all other nodes can be calculated in linear time via a breadth-first search. The worst-case complexity for the standard version of TAGNet is therefore:

$$O\left((\delta + 1) \cdot \left(|V|d^2 + |E|d\right)\right). \tag{24}$$

Of note is that the worst case-complexity is independent of the number of layers. This allows for much deeper propagation.

We further discuss the complexity when utilizing degree messages and a target-specific $\delta$. As noted in Section 3.3, the inclusion of degree messages is equivalent to aggregating an additional edge each iteration. As such, it doesn't effect the model complexity. Furthermore, when utilizing a target-specific $\delta$, an additional $(\delta + 1) \cdot d^2$ operations are added to calculate the attention scores. This is equivalent to updating each one node one additional time and therefore also has no effect on the model complexity.

## B.3  TAGNet + A*Net

We further experiment with combining the pruning strategy of both A*Net and TAGNet. This is achieved by taking the intersection of the edge sets produced by both methods for a node pair $(s, o)$ at iteration $t$. This is because we only want to aggregate an edge if it is not pruned by both methods. For TAGNet, the edge set $\mathcal{C}(s, o, t)$ is defined as in Eq. (6). We further denote the edge set for A*Net as $\mathcal{A}(s, o, t)$. Adapting Eq. (7) we arrive at:

$$\mathbf{x}_q^{(t)}(s, o) = \left( \bigoplus_{(v,r,o) \in \mathcal{C}(s,o,t) \cap \mathcal{A}(s,o,t)} \mathbf{x}_q^{(t-1)}(s, v) \otimes \mathbf{w}_q(v, r, o) \right) \oplus \mathbf{x}_q^{(0)}(s, o). \qquad (25)$$

The performance and efficiency when combining both methods is detailed in Section 4.1 and 4.2, respectively. Lastly, we note that we don't consider combining with the pruning strategy in AdaProp (Zhang et al., 2023) due to its strong similarity with that of A*Net.

## C  Experimental Settings

### C.1  Datasets

We conduct experiments on both the transductive and inductive settings. For the transductive setting, we consider FB15K-237 (Toutanova and Chen, 2015) and WN18RR (Dettmers et al., 2018). For the inductive setting, where the train and test entities are disjoint, we consider the splits generated by Teru et al. (2020) from both FB15K-237 and WN18RR. Four splits are generated from both datasets that vary in size. Of note is that we omit the NELL-995 (Xiong et al., 2017) dataset from both sets of our experiments. This is due to concerns raised by Safavi and Koutra (2020), where they argue that most of the triples in NELL-995 are either meaningless or trivial. The statistics for all the transductive and inductive datasets are given in Tables 5 and 6, respectively.

Table 5: Statistics for Transductive Datasets.

| Statistic | FB15K-237 | WN18RR |
|---|---|---|
| #Entities | 14,541 | 40,943 |
| #Relations | 237 | 11 |
| #Train | 272,115 | 86,835 |
| #Validation | 17,535 | 3,034 |
| #Test | 20,466 | 3,134 |

### C.2  Baselines

In the transductive setting, following (Zhu et al., 2021), we consider a variety of different models. For embedding-based methods we consider TransE (Bordes et al., 2013) (performance from (Nguyen et al., 2018)), DistMult (Yang et al., 2015), ComlEx (Trouillon et al., 2016). For GNN methods we include R-GCN (Schlichtkrull et al., 2018) (performance on WN18RR taken from (Zhu et al., 2021)) and CompGCN (Vashishth et al., 2019). For path-based methods we include DRUM (Sadeghian et al., 2019), NBFNet (Zhu et al., 2021), RED-GNN (Zhang and Yao, 2022), A*Net (Zhu et al., 2022), and AdaProp (Zhang et al., 2023). We note that for AdaProp the original results from (Zhang et al., 2023) utilize 7 and 8 layers for FB15k237 and WN18RR, respectively (see Table 7 in (Zhang et al., 2023)). For other methods such as TAGNet, NBFNet, and A*NET, the number of layers is fixed at 6. To facilitate a fair comparison, we run AdaProp on both datasets using 6 layers. We utilize the official source code [3] and the published hyperparameters.

For the inductive setting, following (Teru et al., 2020; Zhu et al., 2021), we include GraIL (Teru et al., 2020), CoMPILE (Mai et al., 2021), and NeuralLP (Yang et al., 2017) in addition to NBFNet and A*Net.

---

[3] https://github.com/LARS-research/AdaProp

Table 6: Statistics for Inductive Datasets.

| Dataset | | #Relations | Train | | | Validation | | | Test | | |
|---|---|---|---|---|---|---|---|---|---|---|---|
| | | | #Entities | #Query | #Fact | #Entities | #Query | #Fact | #Entities | #Query | #Fact |
| FB15k-237 | v1 | 180 | 1,594 | 4,245 | 4,245 | 1,594 | 489 | 4,245 | 1,093 | 205 | 1,993 |
| | v2 | 200 | 2,608 | 9,739 | 9,739 | 2,608 | 1,166 | 9,739 | 1,660 | 478 | 4,145 |
| | v3 | 215 | 3,668 | 17,986 | 17,986 | 3,668 | 2,194 | 17,986 | 2,501 | 865 | 7,406 |
| | v4 | 219 | 4,707 | 27,203 | 27,203 | 4,707 | 3,352 | 27,203 | 3,051 | 1,424 | 11,714 |
| WN18RR | v1 | 9 | 2,746 | 5,410 | 5,410 | 2,746 | 630 | 5,410 | 922 | 188 | 1,618 |
| | v2 | 10 | 6,954 | 15,262 | 15,262 | 6,954 | 1,838 | 15,262 | 2,757 | 441 | 4,011 |
| | v3 | 11 | 12,078 | 25,901 | 25,901 | 12,078 | 3,097 | 25,901 | 5,084 | 605 | 6,327 |
| | v4 | 9 | 3,861 | 7,940 | 7,940 | 3,861 | 934 | 7,940 | 7,084 | 1,429 | 12,334 |

We note that embedding methods aren't applicable to the inductive setting as the train and test entities are disjoint. For NBFNet, the results on the inductive FB15k-237 splits are reported by us while the results for the WN18RR splits are from Zhu et al. (2022). This is because we observed that we can achieve better performance for NBFNet on the FB15k-237 splits than what was reported in Zhu et al. (2022). Lastly, as with the transductive setting, we run AdaProp with 6 layers to facilitate a fair comparison between it and other path-based GNNs. We also set the hidden dimension to 32 as is with all other path-based GNNs.

### C.3 Evaluation Metrics

In the transductive setting, we report the mean reciprocal rank (MRR), Hits@1, and Hits@10 following the filtered setting as described in (Bordes et al., 2013). For the inductive setting, following (Zhang et al., 2023; Zhu et al., 2022), we only report the Hits@10.

### C.4 Hyperparameter Settings

We list the parameters settings for TAGNet. Under the fixed-$\delta$ formulation it is trained for 20 and 16 epochs on the transductive and inductive setting, respectively. For the specific-$\delta$ formulation, we train for 25 and 20 epochs on the transductive and inductive setting, respectively, as we've found it takes longer to converge. For all transductive and inductive experiments in Table 1 and 2 we set the maximum number of layers to 6 and the hidden dimension to 32. This is to facilitate a fair comparison with NBFNet and A*Net. Furthermore the transductive batch size is fixed at 16. The number of negative samples is tuned from $\{128, 512, 1024, 2048\}$, the dropout from the range $[0, 0.7]$, the learning rate decay from $\{0.9, 0.95, 1\}$, the weight decay from [1e-8, 1e-3], and the adversarial temperature from $\{0.5, 1\}$. For the target specific setting we further test on setting $g$ as its own function or as equal to the score function, $g = f$. We further tune the softmax temperature for attention from $\{0.5, 1, 5\}$. For the inductive setting we further tune the batch size from $\{16, 32, 64, 128\}$ and the learning rate from [1e-4, 1e-2]. Lastly, for all experiments, the offset $\delta$ is tuned from $\{1, 2, 3\}$.

### C.5 Implementation Details

The framework is implemented with PyTorch (Paszke et al., 2019). All experiments were run on a single 32G Tesla V100 GPU. We train TAGNet with the binary cross-entropy loss optimized via the Adam optimizer (Kingma and Ba, 2014). We follow Yang et al. (2017) and augment the graph by including reciprocal edges, such that for an edge $(h, r, t)$, its reciprocal edge $(t, r^{-1}, h)$ is included. In this scenario $r^{-1}$ is considered a distinct relation from $r$.

## D  Additional Analysis on TAGNet

In this section we take a closer look as to what kind of messages are pruned by TAGNet. As noted in Section 3.1 we strive to limit the number of empty and redundant messages. We first analyze how well TAGNet can prune both of those messages. We then examine the reason why some datasets may prune more empty or redundant messages.

We first analyze the number of **empty** and **redundant** messages pruned for both transductive datasets. We report the results in Table 7 as a % of the total number of pruned messages. E.g., For FB15k-237 51% of the total number of pruned messages are empty messages. For simplicity, we limit this study to the the best versions of each model, i.e. $\delta = 2$ for FB15K-237 and $\delta = 3$ for WN18RR. We find that on

FB15k-237, the messages pruned are split evenly between empty and redundant messages. On the other hand, for WN18RR over 90% of the messages pruned are empty messages.

Table 7: % of Messages Pruned that are either Empty or Redundant

| Dataset | % Empty | % Redundant |
|---------|---------|-------------|
| FB15k-237 | 51% | 49% |
| WN18RR | 91% | 9% |

An obvious question is: *Why does the composition of pruned messages differ between datasets?* We believe this can be explained via two properties of each datasets, the density and distance distribution. We measure the sparsity via the mean degree, which is shown in Table 8. We do this as graphs with a low mean degree will contain few connections between nodes, resulting in fewer paths between different nodes and thereby fewer redundant paths. Furthermore, there will be a lower chance of a node visiting another node already on the path, as most nodes are linked to only a handful of nodes. We further show the distance distribution of the test samples, i.e., the % of test samples that are a distance $k$ from each other, in Table 9. This is because when nodes are typically far from each other, the target nodes will aggregate many empty messages. Using Figure 1a as an example, the source and node 7 are a distance 3 from each other. Because of this, in the first two iterations NBFNet will propagate node 6 to node 7, even though node 6 contains no information. However, this is less of an issue between nodes of shorter distances as there fewer iterations needed to reach it. From this, we hypothesize that graphs that feature, on average, a larger distance between nodes will propagate more empty messages.

Table 8: Mean Degree of Transductive Datasets

| Dataset | Mean Degree |
|---------|-------------|
| FB15k-237 | 18.7 |
| WN18RR | 2.1 |

Table 9: Distance Distribution of Test Samples on the Transductive Datasets

| Distance | FB15k-237 | WN18RR |
|----------|-----------|--------|
| 1 | 0% | 35% |
| 2 | 73% | 9% |
| 3 | 26% | 21% |
| 4 | 0.2% | 7% |
| 5 | 0.005% | 9% |
| 6+ | 0% | 18% |

From the results in Table 8 and 9 we make the following observations: **(a)** WN18RR is much sparser than FB15k-237. The higher density of FB15k-237 leads to many more paths and subsequent opportunities to visit a node already on the path. The opposite is true for WN18RR as since the average degree is low, few paths exist in the graph. This results in many more redundant paths existing in FB15k-237 as compared to WN18RR. **(b)** For FB15k-237, the vast majority of test samples are close to each other. This leads to less empty messages. However, for WN18RR the distance covers a much wider range. For example, over 33% of test samples have a distance of 4+ between them. This is only true for 0.205% of samples on FB15k-237. This helps explain why TAGNet mostly prunes empty messages on WN18RR, as the larger distance between nodes leads to many messages that contain no information.