# OpenReview forum: "Distance-Based Propagation for Efficient Knowledge Graph Reasoning"
_EMNLP/2023/Conference — EMNLP 2023 Main_

### Official Review · Reviewer_NkaF · 2023-08-04

**Soundness:** 4

**Excitement:**

3: Ambivalent: It has merits (e.g., it reports state-of-the-art results, the idea is nice), but there are key weaknesses (e.g., it describes incremental work), and it can significantly benefit from another round of revision. However, I won't object to accepting it if my co-reviewers champion it.

**Paper Topic And Main Contributions:**

This paper observes that a limitation of past work on path-based GNN methods for KG completion is (1) Empty Messages and (2) Redundant Messages. To remedy these, the authors present TAGNet, which uses a truncated version of Bellman-Ford to capture paths of a given range of distances (e.g, upper and lower bound). The authors present empirical results that demonstrate that TAGNet achieves state-of-the-art results in transductive settings and good results in inductive setting.

update
Thanks very much for the additional analysis and detailed response. I have increased the soundness score from 3->4.

**Questions For The Authors:**

How should we think about the performance difference of TagNet + A∗Net and the two methods applied independently?

**Reasons To Accept:**

This paper has several strengths:
* **Carefully Targeted Improvements** - The authors nicely short comings of GNN based methods. Then they address these directly and even show that the proposed approach achieves its benefits from these properties (e.g. Table 4). I think this kind of analysis is nice.
* **Empirical Benefits** - The method seems to have empirical advantages compared to past work. I think these gains are nice,
* **Additional methodological twists** - I think that 3.5.1 adds a bit more methodological interest.

**Reasons To Reject:**

I think the paper indeed has several nice contributions. On the other hand concerns include:
* I don't think I really understand how to think about TagNet + A∗Net? I feel the results are somewhat confusing why they are sometimes worse than both methods independently. Perhaps I've missed something here.
* I think understanding the differences in performance across datasets could be improved with some analysis of the underlying graphs.

**Reproducibility:**

3: Could reproduce the results with some difficulty. The settings of parameters are underspecified or subjectively determined; the training/evaluation data are not widely available.

**Reviewer Confidence:**

3: Pretty sure, but there's a chance I missed something. Although I have a good feel for this area in general, I did not carefully check the paper's details, e.g., the math, experimental design, or novelty.

**Typos Grammar Style And Presentation Improvements:**

Line 008: would be nice to say what the efficiency issues are
Line 037: specify recent in GNN.

---

> ### Author Rebuttal · Authors · 2023-08-29
>
> Thank you for your detailed review. We respond to your questions and comments below.
>
> > **Q1**:I don't think I really understand how to think about TagNet + A\*Net? I feel the results are somewhat confusing why they are sometimes worse than both methods independently.
>
> We apologize for the confusion. For the method "TAGNet + A\*Net'', we are actually applying the pruning strategies of both A\*Net and TAGNet (our framework). This works by successively applying the strategies of both methods. We first chose which nodes to propagate to via the A\*Net algorithm. Then, we use the TAGNet algorithm to remove the empty and redundant messages (see Section 3.1 for a detailed discussion on both). We refer to this strategy as **A\*Net + TAGNet** as it combines the strengths of both methods. This is needed as A\*Net is not guaranteed to remove **empty** or **redundant** messages.
>
> In the paper, our method "TAGNet'' is based on NBFNet (no pruning) by default. Therefore, "TAGNet'' in the paper is actually referring to "TAGNet+NBFNet", which applies pruning techniques of "TAGNet'' to NBFNet. "TAGNet + A\*Net'' aggregates fewer messages than "TAGNet''. This is due to the addition of the A\*Net pruning strategy which results in many messages being removed. Since A\*Net' achieves worse performance than NBFNet, it is expected that A\*Net + TAGNet is likely to achieve worse performance than TAGNet. We will update this in the paper to make it clear.
>
>
> Comparing the performance of "A\*Net + TAGNet" and A\*Net, we find that **"A\*Net + TAGNet" performs better than A\*Net on 6/10 datasets**. On the other datasets it achieves comparable performance to A\*Net. This is though "A\*Net + TAGNet" can remove many of the messages propagated by A\*Net. As shown in Table 3 in our paper, A\*Net + TAGNet can reduce the number of messages by over 33\% compared to only A\*Net.
>
> **The strong performance of "A\*Net + TAGNet" relative to A\*Net validates our observations in Section 3.1. As even when removing both empty and redundant messages from A\*Net, the performance is not compromised.**
>
> We will clarify these in the revision.
>
>
> > **Q2**:I think understanding the differences in performance across datasets could be improved with some analysis of the underlying graphs.
>
> We appreciate the suggestion. We take a closer look as to what kind of messages are pruned by TAGNet (i.e., empty or redundant) for the different datasets. We then examine the reason why some datasets may prune more empty or redundant messages.
>
> We first analyze the number of **empty** and **redundant** messages pruned for each dataset. As a reminder, we identified these messages as a main cause of inefficiency in path-based GNNs. TAGNet is meant to address this. We therefore further report which types of messages are more prevalent in the different datasets.
>
> We report the results below as a \% of the total number of pruned messages. E.g., For FB15k-237 51\% of the total number of pruned messages are empty messages. The table shows the analysis for the transductive version of both WN18RR and FB15K-237. For simplicity, we limit this study to the the best versions of each model, i.e. $\delta=2$ for FB15K-237 and $\delta=3$ for WN18RR.
>
> | **Dataset** | **\% Empty** | **\% Redundant** |
> |------------------|-------------------|-----------------------|
> | FB15k-237        | 51\%              | 49\%                  |
> | WN18RR           | 91\%              | 9\%                   |
>
> We find that on FB15k-237, the messages pruned are evenly split between empty and redundant messages. On the other hand, for WN18RR over 90\% of the messages pruned are empty messages. However, **it is not clear why the composition of empty and redundant messages differ so much between datasets**.
>
> To better analyze this, we take a closer look at the sparsity and distance distribution of both datasets. We first analyze the sparsity via the average degree. We do this as graphs with a low average degree will contain few connections between nodes, resulting in fewer paths between different nodes and thereby fewer redundant paths. Furthermore, there will be a lower chance of a node visiting another node already on the path, as most nodes are linked to only a handful of nodes.
>
> With this idea, we present the average degree of FB15k-237 and WN18RR below. We observe that WN18RR is very sparse while FB15k-237. The higher density of FB15k-237 allows many more paths and opportunities to visit a node already on the path. The opposite is true for WN18RR as since the average degree is low, few paths exist in the graph. **This gives us an indication as to why many redundant paths are pruned from FB15k-237 and not WN18RR**.
>
> | **Dataset** | **Avg Degree** |
> |------------------|---------------------|
> | FB15k-237        | 18.7                |
> | WN18RR           | 2.1                 |
>
> **To better understand the number of empty messages pruned**, we further examine the distance distribution of both datasets. This is because when nodes are typically far from each other, the target nodes will aggregate many empty messages. Using Figure 1a as an example, the source and node 7 are a distance 3 from each other. Because of this, in the first two iterations NBFNet will propagate node 6 to node 7, even though node 6 contains no information. However, this is less of an issue between nodes of shorter distances as there fewer iterations needed to reach it. **From this, we hypothesize that graphs that feature, on average, a larger distance between nodes will propagate more empty messages.**
>
> We examine this by calculating the shortest path distance of all test samples. This is shown in the table below. We find that for FB15k-237, the vast majority of samples are close to each other. This limits the potential of empty messages being aggregated. However, for WN18RR the distance covers a much wider range. For example, over 33\% of test samples have a distance of 4+ between them. This is only true for 0.205\% of samples on FB15k-237. This gives further justification as to why most of the pruned messages on WN18RR are empty messages, **as the larger distance between nodes enables TAGNet to prune many messages that contain no information**.
>
> | **Distance** | **FB15k-237** | **WN18RR** |
> |-------------------|--------------------|-----------------|
> | 1                 | 0\%                | 35\%            |
> | 2                 | 73\%               | 9\%             |
> | 3                 | 26\%               | 21\%            |
> | 4                 | 0.2\%              | 7\%             |
> | 5                 | 0.005\%            | 9\%             |
> | 6+                | 0\%                | 18\%            |
>
> We thank the reviewer for their suggestion and will include this analysis in our revision.

---

### Official Review · Reviewer_VWSR · 2023-08-05

**Soundness:** 4

**Excitement:**

3: Ambivalent: It has merits (e.g., it reports state-of-the-art results, the idea is nice), but there are key weaknesses (e.g., it describes incremental work), and it can significantly benefit from another round of revision. However, I won't object to accepting it if my co-reviewers champion it.

**Missing References:**

N/A

**Paper Topic And Main Contributions:**

This work is developed based on NBFNet. The paper point out and analyze the problems of "empty messages" and "redundant messages" in message communication. The author proposes a truncated Bellman-Ford method (TAGNet) to solve the problem. Extensive experiments demonstrate that TAGNet outperforms NBFNet.

**Questions For The Authors:**

See Weaknesses

**Reasons To Accept:**

A. Point out two essential problems in NBFNet, i.e., "empty messages" and "redundant messages"

B. Clear presentation and well-written.

C. Technique solid. The effectiveness of the model is proven both experimentally and theorically.

**Reasons To Reject:**

A. More explanation and experiment comparison between your work and A* Net.

B. Do you follow the principles of establishing bidirectional edges when exploring the shortest path in the WN18RR dataset? Have you considered the paths of duplicate points and edges? I hope this analysis can be more specific.

C. More experiments for the complexity analysis (time and space)

D. Comparison between your work with some anchor-based efficient KGE models, such as Nodepiece [1].

[1] Galkin, M., Denis, E., Wu, J., & Hamilton, W. L. (2021). Nodepiece: Compositional and parameter-efficient representations of large knowledge graphs. arXiv preprint arXiv:2106.12144.

**Reproducibility:**

4: Could mostly reproduce the results, but there may be some variation because of sample variance or minor variations in their interpretation of the protocol or method.

**Reviewer Confidence:**

4: Quite sure. I tried to check the important points carefully. It's unlikely, though conceivable, that I missed something that should affect my ratings.

---

> ### Author Rebuttal · Authors · 2023-08-29
>
> We thank the review for taking the time to help improve our paper. We respond to their questions and comments below.
>
> > **Q1:** More explanation and experiment comparison between your work and A\* Net.
>
> Thank you for your suggestion. We will add such additional discussions and experiments in our revision. We include both below.
>
> We first give a brief explanation of our model vs. A\*Net. Both methods attempt to reduce the number of messages propagated by NBFNet. This is of paramount importance as in practice NBFNet can be very computationally expensive to run.  A\*Net attempts to do so by only propagating messages to a small subset of nodes at each iteration. The nodes are chosen based on a set of learnable parameters. However, we take a new perspective approach to solving this problem. Our framework improves the efficiency by instead not propagating messages we find unnecessary and redundant. These are reasoned from two observations found in Section 3.1. The first is that we want to avoid propagating **empty messages**. These are messages that contain no information, a common problem in path-based GNNs. Secondly, we also want to avoid aggregating paths with too much **redundant information**. This often occurs as some paths may traverse the same nodes or edges multiple times (note: these are strictly walks as they contain repeated nodes). As such, there is no utility in propagating such messages.
>
> However, **it is not guaranteed that A\*Net (or AdaProp) will  remove the empty or redundant messages**. To remedy this, we can apply TAGNet to other path-based GNNs to further improve their efficiency. This is because TAGNet is a general framework that can work in conjunction with any path-based GNN. In our paper we show the results when combining the pruning strategies of TAGNet and A\*Net. We refer to this as ``A\*Net + TAGNet".  We observe that when using both methods, over 33\% of the messages propagated by A\*Net can be removed (see Table 3 in our paper). This proves the need for TAGNet, even when using efficiency-minded methods like A\*Net. Furthermore, as shown in Tables 1 and 2,  **when pruning messages via both of our strategies the performance is the same or better than only A\*Net**. This means we can greatly improve the efficiency of A\*Net with no loss in performance.
>
> To further compare TAGNet and A\*Net, we also show the number of messages propagated by both methods **individually** and  **combined**. We further include a comparison with NBFNet. The results are shown in the Table below for FB15k-237 when varying the number of layers from 6, 7, and 8. For TAGNet, we use the best performing $\delta$ (i.e., $\delta=2$).
>
> We observe that while A\*Net may propagate less messages **by itself**, it still contains many empty and redundant messages. This can be seen as A\*Net+TAGNet prunes a large percentage of the messages propagated by just A\*Net. This is further amplified as the number of layers increases.  Most importantly we find the performance between A\*Net and TAGNet on all datasets are comparable and better than just A\*Net (see Tables 1 and 2 in paper). **This emphasizes the need for TAGNet even when using A\*Net, as it can help remove unnecessary and potentially harmful messages**.
>
> We appreciate these suggestions and we will revise our paper with the explanation and experiment comparison.
>
> | **Method** | **Layers=6** | **Layers=7** | **Layers=8** |
> |-----------------|-------------------|-------------------|-------------------|
> | NBFNet          | 3,265,380         | 3,809,610         | 4,353,840         |
> | A\*Net          | 230,638           | 284,198           |   336,523              |
> | TAGNet          | 1,626,144         | 1,627,239         | 1,627,397         |
> | A\*Net + TAGNet | 84,237            | 138,581           | 139,249           |
>
>
> > **Q2**: Do you follow the principles of establishing bidirectional edges when exploring the shortest path in the WN18RR dataset?
>
> Yes, we include all bidirectional edges when computing the shortest path. It is calculated via a BFS, which visits each node or edge in the graph at most once. We will include this in our revision.
>
> > **Q3**: Have you considered the paths of duplicate points and edges?
>
> In fact, **this is one of the motivations and advantages of our method**. In our study we observe that methods such as NBFNet tend to aggregate paths with duplicates edges and nodes (see observation 2 in Section 3.1). We give an example using Figure 1a in our paper when running NBFNet with 6 layers. In the first two iterations node 4 aggregates two paths including (source, 4) and (source, 3, 4). However in the 5th iteration, node 4 aggregates paths of length 5 that contain redundant nodes. An example would be the path (source, 4, source, 4, source, 4), where we go back and forth between the source and node 4.
>
> This observation is one of the guiding points of our model design. To combat excessive backtracking, we introduce a hyperparameter $\delta$. Formally, $\delta$ is an offset that defines the maximum path length that nodes can aggregate. I.e., when $\delta=1$, we allow nodes to aggregate all paths of length up to and including $\text{shortest-path}(u, v) + 1$. The hyperparameter $\delta$ also allows us to indirectly control the amount of duplicate edges/nodes we aggregate. Going back to Figure 1a, when $\delta=4$, node 4 will additionally aggregate the path (source, 4, 6, 7, 6, 4). This allows node 4 to gather information from node 7. However if $\delta=3$, this path won't be aggregated.
>
> The decision of whether to aggregate redundant messages depends on the dataset. Since $\delta$ is a hyperparameter it can be customized for each dataset. Furthermore, we also introduce a target-specific $\delta$ in Section 3.5. This allows further customization down to the sample level.
>
> > **Q4**: More experiments for the complexity analysis (time+space)
>
> The runtime on both FB15k-237 and WN18RR are shown below for each method. We used the same batch size for each method and chose $\delta=2$ for TAGNet. All experiments were run on a single NVIDIA RTX A5000. **The runtime represents the number of minutes needed to train 1 epoch**.
>
>
> | **Method** | **WN18RR** | **FB15k-237** |
> |-----------------|-----------------|-----------------|
> | NBFNet          | 16.0            | 48.1            |
> | TAGNet          | 10.8            | 34.4            |
> | A\*Net          | 8.4             | 33.7            |
> | TAGNet + A\*Net | 5.6             | 26.3            |
>
> We find that both TAGNet and A\*Net can improve the efficiency of NBFNet.
> We note that A\*Net is slightly faster than TAGNet. This is as expected as it tends to remove many more messages than A\*Net. However, since A\*Net and TAGNet are **complementary** and **not competing methods**, combining them (i.e., TAGNet + A\*Net) can further improve the efficiency.
> Furthermore, when using TAGNet in conjunction with A\*Net we are still able to significantly reduce the number of messages relative to A\*Net. This is demonstrated in Table 3. Lastly, we note that TAGNet achieves strong performance even when improving the efficiency. As shown in Table 1, TAGNet can outperform even NBFNet on the transductive datasets.
>
>
> > **Q5**: Comparison between your work with some anchor-based efficient KGE models, such as Nodepiece
>
> While both papers are concerned with efficiency, we respectively point out that **their scopes are substantially different**. NodePiece is concerned with reducing the number of parameters for **KG embedding models (KGE)**. As opposed to standard KGEs that represent each node with one embedding, NodePiece constructs a node's representation from a series of anchor nodes and relational contexts. On the other hand, our work is concerned with improving efficiency of **path-based GNNs** by reducing the number of messages aggregated. Since path-based GNNs don't feature node embeddings, they often have a much smaller number of parameters than KG embedding models but they usually face the efficiency issue that motivates our current work.
>
> Furthermore, we note that the efficiency of NodePiece comes at a significant cost to performance. In their study, they are able to achieve a MRR of 0.256 and 0.403 on FB15k-237 and WN18RR, respectively. This is considerably lower than our performance of 0.421 and 0.562 on FB15k-237 and WN18RR, respectively.
>
> We will revise our paper to include a discussion of NodePiece in the related work.

---

### Official Review · Reviewer_jKiL · 2023-08-05

**Soundness:** 4

**Excitement:**

3: Ambivalent: It has merits (e.g., it reports state-of-the-art results, the idea is nice), but there are key weaknesses (e.g., it describes incremental work), and it can significantly benefit from another round of revision. However, I won't object to accepting it if my co-reviewers champion it.

**Paper Topic And Main Contributions:**

Link prediction approaches aim to address the problem of knowledge graph incompleteness. The methods that leverage aggregation of path information through a learning process have recently shown promising performance in link prediction tasks. In particular, the neural Bellman-Ford method presents a neural version of the Bellman-Ford algorithm that computes path information via dynamic programming. The aim is to learn pairwise node representation inductively.
The paper points out that the neural Bellman-Ford algorithm suffers from inefficiency due to two reasons: a) Empty message: the nodes which are far from the source node receive empty messages when their shortest path to the source node is bigger than the number of propagation layers, b) Redundant message: For a large number of layers, the close nodes get most of the information from the shortest path propagation steps, and longer paths do not give substantial information while presenting a more complexity.
By proposing a modification over existing works, the paper aims to address the mentioned issues to enhance the efficiency of path-based methods. In particular, a new path-based GNN model is proposed by revising the formulation of the NBF model to only consider the iteration t between the shortest path with the offset of \delta. As \delta is a hyperparameter, the paper further proposes an attention mechanism to pay attention to the most relevant representation from 1 to \delta offsets.
The experiments have been done on both transductive and inductive settings on popular benchmark datasets such as FB237 and WN18RR. The proposed method obtains competitive performance to NBFnet and A*-Net while reducing the number of propagated messages.

**Questions For The Authors:**

Question A: What is the running time of the proposed approach and how does it reduce compared to the other competitors?
Question B: What is the complexity of computing the shortest distance for each node pair? How is it scalable?

**Reasons To Accept:**

-- The paper points out an overlooked problem in recent state-of-the-art path-based link prediction methods and presents a way to address this problem.

-- The proposed method preserves the performance while reducing the number of messages, addressing efficiency issues of previous works.

**Reasons To Reject:**

-- The writing and presentation of the paper could be further improved to be more understandable for various readers with different backgrounds, e.g., equation 1 and its description are not enough to understand them well, as the notation and operators are not defined well and it is required to look at the references to find its actual meaning while it is expected that the paper is standalone, or similarly indicator and u, v, msg are not defined well. I have to read the NBFNet paper first to understand them.

-- The central aspect of the suggested approach involves a straightforward adjustment to existing research, with a limited technical contribution.

-- It is expected to have running time besides time complexity analysis. Moreover, a discussion about running time and time complexity should be in the main body of the paper as it is one important aspect of the paper.

-- Theorem 1 looks somehow trivial.
Adding target-specific \delta does not work according to the overall results.

**Reproducibility:**

4: Could mostly reproduce the results, but there may be some variation because of sample variance or minor variations in their interpretation of the protocol or method.

**Reviewer Confidence:**

3: Pretty sure, but there's a chance I missed something. Although I have a good feel for this area in general, I did not carefully check the paper's details, e.g., the math, experimental design, or novelty.

**Typos Grammar Style And Presentation Improvements:**

Page 4, line 309:C ->  \mathcal{c}
Page 5, line 338: given -> give
In An Illustrative Example, it might be better to first determine s and o, and then run the process.
Page 5, line 383: no verb is given

---

> ### Author Rebuttal · Authors · 2023-08-29
>
> Thank you for taking the time to review our paper. We respond to your comments below.
>
> > **Q1**: "The writing and presentation of the paper could be further improved to be more understandable for various readers with different backgrounds."
>
> Thank you for your suggestion. We hope to make our paper as accessible as possible to those from different backgrounds. To achieve this goal, we plan to give a more in-depth definition of the background equations in Section 2 below. We further define the different notations and operators used in the paper. We will revise our paper to more properly define these equations and other notations.
>
> **Equation 1**:
>
> $$
> h_q(s, o) = \bigoplus_{t=1}^T \bigoplus_{p \in P_{s,o}^t} \bigotimes_{i=1}^{\lvert p \rvert} \mathbf{w}_q(e_i)
> $$
>
> We are interested in whether there exists and edge $(s, q, o)$ where $s$ and $o$ are nodes and $q$ is a relation. Eq. 1 solves this by aggregating all paths between $s$ and $o$. Since this can be too computationally expensive, we limit to aggregating all paths up to a max length $T$. The set of all paths of length $t$ between $s$ and $o$ is given by $p \in P_{s,o}^t$. A single path $p$ is characterized by all the edges in that path $e_i \in p$. We assume that for each edge $e_i$ there exists an edge representation $\mathbf{w}_q(e_i)$ that is conditional on the query relation $q$ we are trying to predict.
> The operators $\bigoplus$ and $\bigotimes$ can be represented by the sum, max, min, or multiplication. In Zhu et al. (2021) they show that this formulation can capture many existing graph algorithms including the katz index, personalized pagerank, the graph distance, and more.
>
> **Equations 2 and 3**:
>
> $$
> h_q^{(0)} (s, o) = \mathbf{1}_q (s = o)
> $$
>
> $$
> h_q^{(t)} (s, o) = \bigg(\bigoplus_{(x, r, o) \in \mathcal{E}(o)} \mathbf{h}_q^{(t-1)}(s, x) \otimes \mathbf{w}_q(x, r, o) \bigg)   \oplus\mathbf{h}_q^{(0)} (s, o),
> $$
>
> An issue with Equation 1 is that even when $T$ is small, it can be very inefficient to consider all possible paths. To combat this problem, the generalized Bellman-Ford algorithm is used. This algorithm works in an iterative manner. For the edge we are trying to predict, $(s, q, o)$, we first initialize its edge representation via the indicator function in Eq. 2. This evaluates to the {**1** vector when the edge $s = 0$ and otherwise is **0**. As before, we also associate an edge representation for each edge, denoted as $\mathbf{w}_q(x, r, o)$.
>
> The iterative computation in Eq. 3 has two components. In the first, we consider all edges where $o$ is the object entity. They are combined via an aggregation function $\bigoplus$ (e.g., sum, max, etc.). For each edge $(x, r, o)$ we consider the path information from $s$ to $x$, $\mathbf{h}_q^{(t-1)}(s, x)$, and the edge representation from $x$ to $o$. Since the pair representations are initialized to **0**, at $t=1$ there is no path information from $s$ to $x$. As such only the edge representation $\mathbf{w}_q(x, r, o) \geq 0$. Therefore we only consider paths of length 1. At $t=2$ the representations $\mathbf{h}^{(1)}(s, x)$ will not be non-zero. This will represent the cost of traversing the paths of length 1 between $s$ and $x$. Thereby aggregating $\mathbf{h}^{(1)}(s, x)$ and $\mathbf{w}(x, r, o)$, we are considering the cost from $s$ to $x$ and $x$ to $o$. This is a path of length 2. This is done until some maximum iteration $T$.
>
> The second component $\mathbf{h}_q^{(0)} (s, o)$ is the boundary condition. If $s = 0$ we know the shortest path between them must be 0. This equation degenerates to the shortest-path Bellman-Ford when $\otimes=+$ and $\oplus=min$.
>
> **Equation 4**
>
> $$
> h_o^{(0)} = \text{INDICATOR}(u, v, q)
> $$
>
> $$
> h_o^{(t)} = \text{AGG} \Big( \\{ \text{MSG} (\mathbf{h}_x^{(t-1)}, \mathbf{w}_q(r)) \, | (x, r, o) \in \mathcal{E}(o) \\} \cup  \\{ \mathbf{h}_o^{(0)} \\} \Big)
> $$
>
> Zhu et al. (2021) introduce a neural version of the generalized bellman ford algorithm (Eq. 2 and 3) . This is achieved via the introduction of learnable parameters. This includes:
> - Instead of the **1** vector, the INDICATOR function assigns a learnable embedding when the condition is true. Otherwise it assigns the **0** vector.
> - A learnable embedding for each relation $r$. As such, $\mathbf{w}_q(r)$ is not conditional on $q$ and can be written as $\mathbf{w}(r)$.
> - A MSG function for combining the representation of node $x$ and the relation embedding, i.e., $\mathbf{h}_x^{(t-1)}$ and $\mathbf{w}(r)$. This replaces $\otimes$. An example is the elementwise product.
> - Instead of the $\bigoplus$ operator, an AGG function is used. An example is the summation. This also includes a learnable projection matrix $W$. In practice, NBFNet utilizes the PNA aggregation (Corso et al., 2020).
>
> > **Q2**: "straightforward adjustment to existing research. Limited technical contribution."
>
> We respectfully disagree this comment. In the following, we show that our work **identifies and solves two key efficiency issues plaguing all path-based GNNs**. Then we demonstrate that it provides a new perspective to improve the efficiency which are complementary to existing efficiency strategies.
>
> These two issues are:
> - **Empty Messages**: Current path-based GNNs propagate many empty messages. These are messages that contain no information, a common problem in path-based GNNs. We give an example from Figure 1a in our paper. At iteration 1, node 6 contains no information (i.e., it was initialized to $\mathbf{0}$). However, it is still aggregated by node 4 and node 7. **This needlessly adds additional computation without resulting in node 6 containing more path information.**
> - **Redundant Information**: Path-based GNNs are liable to aggregate paths that contain a lot of redundant information. This often occurs as some paths (note: strictly speaking they are walks not paths) may traverse the same nodes or edges multiple times. We give an example from Figure 1a when using NBFNet. In the first two iterations, node 4 aggregates two paths including (source, 4) and (source, 3, 4). However, in the 5th iteration, node 4 aggregates paths of length 5 that contain redundant nodes. For example, the path (source, 4, source, 4, source, 4), where we go back and forth between the source and node 4. **Such paths add little to no additional path information that wasn't already aggregated by previous messages**.
>
> While methods like A\*Net do improve the efficiency of path-based GNNs, **they may still propagate empty or redundant messages**. Since in A\*Net the nodes propagated to are selected via a set of learnable parameters, there's no guarantee it'll learn to remove the empty and redundant messages. We demonstrate this in Table 3 in our paper. We calculate the number of messages propagated when using  just A\*Net and A\*Net + TAGNet (our method). We observe that when using both methods, over 33\% of the messages propagated by A\*Net can be removed. Furthermore, we can also improve on the time of both NBFNet and A\*Net when using TAGNet (see the Table in Q3). This proves the need for TAGNet. In addition, as shown in Tables 1 and 2, when pruning messages via both of our strategies **the performance is the same or better than only A\*Net**.
>
> This means we can greatly improve the efficiency of A\*Net with no loss in performance. The same can be applicable for AdaProp, which is similar to A\*Net. We further emphasize that these gains in efficiency are on top of those already gained on just NBFNet, where we can remove over 90\% of the messages (see Figure 3 in our paper). As such, **TAGNet can be added to any path-based GNN to improve the efficiency while achieving comparable or better performance**.
>
> >**Q3**:  " What is the running time of the proposed approach and how does it reduce compared to the other competitors?"
>
> The runtime on both FB15k-237 and WN18RR are shown below for each method. We used the same batch size for each method and chose $\delta=2$ for TAGNet. All experiments were run on a single NVIDIA RTX A5000. **The runtime represents the number of minutes needed to train 1 epoch**.
>
>
> | **Method** | **WN18RR** | **FB15k-237** |
> |-----------------|-----------------|-----------------|
> | NBFNet          | 16.0            | 48.1            |
> | TAGNet          | 10.8            | 34.4            |
> | A\*Net          | 8.4             | 33.7            |
> | TAGNet + A\*Net | 5.6             | 26.3            |
>
> We find that both TAGNet and A\*Net can improve the efficiency of NBFNet.
> We note that A\*Net is slightly faster than TAGNet. This is as expected as it tends to remove many more messages than A\*Net. However, since A\*Net and TAGNet are **complementary** and **not competing methods**, combining them (i.e., TAGNet + A\*Net) can further improve the efficiency.
> Furthermore, when using TAGNet in conjunction with A\*Net we are still able to significantly reduce the number of messages relative to A\*Net. This is demonstrated in Table 3. Lastly, we note that TAGNet achieves strong performance even when improving the efficiency. As shown in Table 1, TAGNet can outperform even NBFNet on the transductive datasets.
>
> >**Q4**:  "Theorem 1 looks somehow trivial."
>
> Our goal with Theorem 1 is to prove that the truncated Bellman-Ford algorithm (see Eqs. 6 and 7) can capture the same information as in Eq. 5. As a reminder, Eq. 5 demonstrates the information we hope for our algorithm to capture for a node pair (s, o). This is all paths of length in the range [dist(s, o), dist(s, o) + $\delta$]. Furthermore, not only do we want to aggregate this information, but **we want this to be the only information we aggregate**. We do not want to aggregate any paths of length greater than dist(s, o) +$\delta$. Therefore, in addition to our empirical results, **the theorem is necessary** to provide theoretical evidence to support the advantages of our method.
>
> Meanwhile, we also face challenges to prove this as all target nodes are being updated in parallel. That is, given a source $s$, all target nodes $o \in \mathcal{V}$ are also updated. This is due to the iterative nature of the algorithm. As such, we are faced with the challenge of ensuring we aggregate the paths of the correct length range for the different target nodes.
>
> We prove that our algorithm can achieve this. Specifically, Theorem 3.1 proves that our formulation (see Eqs. 6 and 7) does in fact satisfy Eq. 5. This is achieved by only aggregating the nodes in the set C(s, o, t) (see Eq. 6). This leads us to  only aggregate paths of length in the range [dist(s, o), dist(s, o) +$\delta$]. As such the final representation of the node pair (s, o), denoted by $x_{(s, o)}^F$
> , only contains this path information. Please see Appendix A.1 for a detailed proof for this theorem.
>
> > **Q5**: "target-specific $\delta$ does not work"
>
> We respectfully point out that the target-specific $\delta$ leads to an increase in performance on **all transductive and inductive versions** of the WN18RR dataset. Furthermore, the performance gains on some splits like WN18RR v1 and v3 can be significant with an increase in MRR of around 0.01.
>
> However, we admit that there is limited benefit to using a target-specific $\delta$ on FB15k-237. This is likely caused by the different characteristics of the two graphs. Mainly, that FB15k-237 is much denser than WN18RR. For example, on the transductive version of FB15k-237, 99.8\% of test samples have a shortest path distance of 2 or 3. However, for WN18RR the shortest path distances varies greatly, with about 33\% of the test samples having a shortest path distance of $\geq 4$. Please refer to Figure 1 in our paper for the full distribution.
>
> The low variation in the shortest path length implies that a target-specific $\delta$ will have a limited impact. This is because there is little difference in how any two nodes in the graph relate to each other. So one value of $\delta$ will likely work well for most node pairs. On the other hand, for WN18RR the distance between nodes can vary greatly. As such, some node pairs may have many shorter paths connecting them while other may have few longer paths. In this case, tailoring the $\delta$ to each sample may be desirable.
>
> Lastly, we note that in general, there is a strong incentive to use a target-specific $\delta$ as it provides additional flexibility. Instead of having one fixed $\delta$, it allows us to customize the path information to each sample. This can be very useful for datasets where paths of different length relative to the shortest path distance may vary greatly in importance. Furthermore, it simplifies the tuning process, as only a maximum $\delta$ needs to be specified. Lastly, we note that it has no effect on the complexity as shown in Appendix B.2.
>
> > **Q6**:  "What is the complexity of computing the shortest distance for each node pair? How is it scalable?"
>
> We use a BFS. For a single sample the worst case is O($\lvert E \rvert + \lvert V \rvert$) where $\lvert E \rvert$ and $\lvert V \rvert$ are the number of edges and nodes in the graph, respectively. However, we note that this can be computed before training and stored on disk. As such, this can be considered an offline operation, **thereby not effecting the training and testing runtime**.
>
> > **Q7**:  Grammar+Writing
>
> Thank you for your comments. We will revise our paper accordingly.

---

### Meta-Review · Area_Chair_Ps9Y · 2023-09-17

**Recommendation:** 4

**Metareview:**

This paper presents TAGNet, a new model for knowledge graph completion which addresses the limitations of a previous model that propagates “empty messages” and “redundant messages”. This is achieved by only aggregating paths in a fixed window for each source-target pair. The model is evaluated on two standard knowledge graph completion datasets (FB15k-237 and WN18RR).

The reviewers agree that the paper is sound. They also agree that the proposed model is carefully designed to deal with the presented limitations of previous models (such as NBFNet and A*Net). Furthermore, the model appears to be empirically stronger than previous methods. However, the reviewers raised concerns about the time complexity of the model and asked for a comparison of the running time of TAGNet with that of the other models. Those concerns were partially addressed in the response. Finally, the reviewers also raised concerns about the significance of the contribution and the lack of novelty with respect to NBFNet.

Overall, it is my opinion that the work is interesting and technically solid. I also agree with various concerns that were raised in the reviews, but I think that these concerns are minor.

---

### Decision · Program_Chairs · 2023-10-07

**Decision:**

Accept-Main

**Comment:**

This paper presents TAGNet, a new model for knowledge graph completion which addresses the limitations of a previous model that propagates “empty messages” and “redundant messages”. This is achieved by only aggregating paths in a fixed window for each source-target pair. The model is evaluated on two standard knowledge graph completion datasets (FB15k-237 and WN18RR).

The reviewers agree that the paper is sound. They also agree that the proposed model is carefully designed to deal with the presented limitations of previous models (such as NBFNet and A*Net). Furthermore, the model appears to be empirically stronger than previous methods. However, the reviewers raised concerns about the time complexity of the model and asked for a comparison of the running time of TAGNet with that of the other models. Those concerns were partially addressed in the response. Finally, the reviewers also raised concerns about the significance of the contribution and the lack of novelty with respect to NBFNet.

Overall, it is my opinion that the work is interesting and technically solid. I also agree with various concerns that were raised in the reviews, but I think that these concerns are minor.